# Race (black-white) and sex inequalities in tooth loss: A population-based study

**Lívia Helena Terra e Souza**[1]*, **Fredi Alexander Diaz-Quijano**[2], **Marilisa Berti de Azevedo Barros**[1], **Margareth Guimarães Lima**[1]

**1** Department of Collective Health, Collaborating Center for Health Situation Analysis (CCAS), School of Medical Science, University of Campinas, Campinas, São Paulo, Brazil, **2** Department of Epidemiology, Laboratory of Causal Inference in Epidemiology [Laboratório de Inferência Causal em Epidemiologia], School of Public Health, University of São Paulo, São Paulo, Brazil

* liviahelenaterra@gmail.com

**Data Availability Statement:** The data underlying the results presented in the study are available from Unicamp Research Data Repository, V1 https://doi.org/10.25824/redu/J9YOTW, Souza,

## Abstract

The effect of health inequalities is determined by different socioeconomic, sex, and race conditions. This study aimed to analyze the association of tooth loss with race (defined by self-reported skin color) and sex. Based on the hypothesis that the association between tooth loss and race may be modified by sex, we also aimed to evaluate possible interactions between race and sex in association with this event, in a population-based study in the city of Campinas, Brazil. A directed acyclic graph was used to select covariates. The prevalence, of tooth loss was 19% higher in black women compared to white men (Prevalence ratio [PR]: 1.19; 95%CI: 1.05–1.34). Moreover, the prevalence of tooth loss in black women was 26% higher than in white women (PR: 1.26; 95%CI: 1.13–1.42); and, within the strata of black people, black women had 14% higher dental loss (PR: 1.14; 95%CI: 1.02–1.27) compared to black men. This study found a significant interaction between race and sex in tooth loss, with a disadvantage for black women. In addition, this work contributes to the discussion of health inequities and can support policies for the provision of universal dental care.

## Introduction

The impact that social inequalities place on health is often evidenced in research and academic discussions [1, 2]. The effect of these inequalities crosses several dimensions of health and is determined by different conditions, including socioeconomic [3, 4], sex [5], and race disparities [6]. Socioeconomic status is a strong consequence of racial inequalities [7–9], since blacks individuals have a lower educational level, lower income, and tend to live in places of high social vulnerability [10, 11].

In addition to economic issues, it is necessary to consider other disadvantages, which tend to remain in various dimensions of life, even after black slavery was abolished in Brazil, which lasted about 300 years. Racial minorities, in this case black people, may biologically embody the effects of racism [7, 12], with every day (or even less common) discriminatory exposures [12]. Adversities throughout life, such as poverty, psychosocial stresses, stereotypes, and the context of housing, can affect the physical and mental health, altering cardiocirculatory,

Lívia Helena Terra e;Diaz-Quijano, Fredi Alexander; Barros, Marilisa Berti de Azevedo;Lima, Margareth Guimarães, 2022, "Replication data for: race (black-white) and sex inequalities in tooth loss".

**Funding:** Funding information: FADQ is beneficiary of a fellowship for research productivity from the National Council for Scientific and Technological Development - CNPq, process/contract identification: 312656/2019-0. The National Council of Technological and Scientific Development (CNPq) funded the research (Grant number: 309073/2015-4) and funds the productivity scholarship granted to MBAB. The survey was supported by the State of São Paulo Research Assistance Foundation (FAPESP) (Process number: 2012/23324-3), Ministry of Health and the Secretary of Health of Campinas (Partnership UNICAMP/Funcamp/SMS 136/14). Funders had no role at any stage in the conduct of the manuscript. These authors contributed equally to this work.

**Competing interests:** The authors have declared that no competing interests exist.

metabolic and immunological functions [8]. In oral health is possible that inequities are due to poverty [10], levels of education [4] or discrimination in health care [13]. In Brazil, skin color has been studied as proxy of race and this indicator is commonly referred as color/race [14–16]. In health, in contrast to the only biologic approach, the race may be considered as a concept socially constructed by historic dynamics and power relations [8, 12].

The literature has pointed out worse oral health conditions among black individuals compared to whites. The highest prevalence, among black individuals, of untreated caries [17, 18], periodontal diseases [19], greater need for prosthesis, difficulty in accessing dentists [6], and the influence of race on the decision of dental mutilation [13] are examples of racial inequalities in oral health in Brazil. Race is associated with the difficulty in accessing health care [20], especially oral health care. Since it is a service with higher costs due to the high prevalence and recurrent cumulative nature of caries and periodontal disease, the oral cavity is even considered the most expensive part of the body to treat [21, 22]. In this sense, socioeconomic level and monthly income, for which black individuals are highly vulnerable [17], are determining factors in the use of oral health services [4, 23].

Tooth loss is among the conditions that have most impacted the health of the world population in the past two decades. In other countries, ethnic differences were found in a sample in East London, with disadvantages for black individuals for loss of posterior tooth [24], and in Brazil, evidence of more frequent tooth loss was found in black individuals when compared to white individuals [6, 25].

Attention to sex inequalities is also required, considering the historical aspects of oppression of women, which persist today, especially regarding the situation of work, income, double shifts, and violence [26, 27]. These issues seem to have health effects, especially on emotional aspects [27], but a higher occurrence of diseases [27] and worse self-rated health [3] has also been shown for women. Nevertheless, in other dimensions, such as worse health risk behaviors and early mortality, the scenario is unfavorable for men [5]. Thus, women seem to be more vulnerable to health conditions that are more limiting and chronic, while men present more behavioral and lethal problems. Research has shown an important impact on oral conditions due to the fact of being female [28–30]. Other studies have found a disadvantage for women concerning chewing, communication, and pain difficulties [31], negative self-assessment of oral health [32], and tooth loss [3]. However, there is no consensus in the literature, and there is one study reporting the absence of gender differences [33].

The evidence of racial and sex inequalities in tooth loss could bring the subgroup of black women underprivileged in relation to this theme. In this way, it is worth considering the analytical strategy of intersectionality, developed by Crenshaw in 1989 to study the black feminism. Intersectionality, drawing on a set of social issues and perspectives, explains an experience of multiple subordination that cannot be reduced to the simple sum of underprivileged conditions [34].

This study aimed to investigate a possible effect and interactions between race and sex on tooth loss, in a population-based study. To our knowledge, there are no studies looking at the relationship between race and sex in relation to tooth loss and this study contributes to filling this gap.

## Methods

### Study design and sampling

This research is a cross-sectional, population-based study, conducted with data from the Health Survey of the Municipality of Campinas (ISACamp) in Brazil.

A stratified probabilistic sampling was carried out by clusters and in two stages: census tract and household. In the first stage, 70 census tracts were drawn with probability proportional to

the size (number of households). 14 census sectors were drawn from each of the five existing Health Districts in the municipality. In the second stage, households were selected by a systematic draw applied to the list of households existing in each of the sectors drawn [35].

A minimum sample size of 1000 individuals was defined for the age domains of adolescents (10 to 19 years old) and elderly (60 years or over) and 1400 for adults (20 to 59 years old). To reach this sample size, after updating the address list of the selected sectors in the field, the numbers of households for each age domain were independently selected. The definition of the number of households that should be drawn was based on the expected average number of people per household (people/household ratio) for each district, based on the 2010 Census data. The desired sample size (1000 or 1400 people) was divided by the corresponding people/household ratio. However, considering the presence of non-responses, larger numbers of households were drawn, considering non-response rates from previous surveys. Data were collected from 3021 people aged 10 or over [36], using a pre-coded questionnaire, applied through tablets, by interviewers trained.

### Study variables

The outcome variable was self-referred tooth loss, extracted of the question: "Have you ever lost a tooth (upper or lower)? If so, did he lose one, more than one or all of his teeth? (Disregarding extracted teeth to place braces, wisdom and milk teeth)", with the options to response: (1) No, (2) yes, only one tooth, (3) yes, more than one tooth, (4) yes, all teeth. The variable was analyzed in two ways: 1) as a dichotomous variable (loss of at least one tooth or none); and 2) as an ordinal variable classifying each participant into any of the following four categories: No loss, loss of 1 tooth, loss of more than one tooth (but not all), or loss of all teeth. When asking about tooth loss, the interviewers informed the respondents to exclude teeth extracted for orthodontic reasons, disregarding teeth extracted for braces, wisdom and baby teeth. Tooth loss is an important marker of oral health, since each tooth is a dental organ and its subtraction has consequences on occlusal adjustment, temporomandibular joint disorder, reduction of the masticatory board and consequently overload for the digestive system [37]. Besides, the loss of at least one tooth since it represents neglect in the dental field, resulting in an increase in the level of severity of oral diseases, in addition to reflecting the model of oral health care adopted and the way individuals understand the condition [38]. Therefore, we decided to focus the primary analysis on the dichotomous outcome, which also represents the most practical way to assess and present interactions [39, 40]. However, on the other hand, the analysis of the same variable as ordinal can show us a gradient of tooth loss [41].

The race variable studied was self-reported. In Brazil this is the official method of racial classification since 1991, based on the individuals' own perception of the color of their skin. In the present study, race/skin color variable was denominated "race" and categorized as black, brown, and white individuals. It was verified that the categories of black and brown presented a similar association for tooth loss, in each sex and in each age, so these two categories were joined to compose the group of black individuals (also used by IBGE-Brazilian Institute of Geography and Statistics) [42]. Individuals who declared themselves as yellow, indigenous, or other races were not studied, as they represented less than 2% of the population.

In this research, the application of directed acyclic graphs (DAG) was used [43] to identify the factors that would need to be conditioned to control confounding. DAGs have been increasingly used as a tool to guide the design and analysis of epidemiological studies [43]. These graphshelp guide the adjustment to a specific set of variables, avoiding problems such as collision bias, overfitting, and unnecessary adjustments [44, 45].

The DAG included the following measured socioeconomic and demographic variables: age, schooling in years of study (0–4, 5–8, and 9 or more years), monthly income per capita in minimum wages (MW), district of residence (East, North, Northwest, Southwest, and South), last visit to the dentist (less than 6 months, between 6 months and less than 1 year, between 1 year and less than 2 years, and 2 years or more or never consulted). Other variables such as, health behavior, patterns of participation (individuals who answered the survey because they spend more time at home, such as women, the elderly, and children), and familiarity (characteristics of the family, such as housing and cultural regions), were not measured in the study were included in the DAG. to assess the need to conditioning on them in the multiple models carried out.

The Vulnerability Index of the State of São Paulo (IPVS) was also included, referring to each sector included in the survey, stratified as the most very low, very low, low, average, high, and very high vulnerability.

The IPVS consists of a typology of situations of exposure to vulnerability, adding to the income indicators others referring to the family life cycle and schooling, in the intra-urban space. It is composed of socioeconomic variables (income and literacy status), as well as demographic variables related to the family life cycle (presence of younger children, age and sex of the head of the family). The IPVS is conceptually composed of two assumptions, the first being the finding that the numerous dimensions of poverty need to be considered in a study on social vulnerability. In this sense, the IPVS operationalizes the concept of social vulnerability [46] that the vulnerability of an individual, family or social group refers to their greater or lesser capacity to control the forces that affect their well-being. Thus, vulnerability to poverty is not limited to considering income deprivation, but also family composition, health conditions and access to medical services, access and quality of the educational system, the possibility of obtaining quality work and adequate remuneration, the existence of legal and political guarantees. The second assumption on which the IPVS is based is the consideration that spatial segregation is a phenomenon present in State of São Paulo's urban centers and that it contributes decisively to the permanence of patterns of social inequality.

In the DAG, we represented the effects of race and sex on oral health, which can be mediated by income [3, 4], education [4] and, more proximally to the outcome, by health behaviors and the dental care [6]. Other variables, such as age, familiarity, patterns of participation in the research, place of residence and IPVS [7], were also considered in the causal diagram (Fig 1). Because it can be a list of unmeasured common causes of both place and race, we follow a convention that visually simplifies the DAG by representing all unmeasured variables with the same causal structure (i.e., the same arrows in and out) as a single node [47]. In this case, we included the concept of "familiarity" as that node representing all common determinants of both, place of residence and race. On the other hand, vulnerability (as a context variable measured with the IPVS) in which people are born, would neither determine or be determined by race. But it would be determined by factors such as familiarity and place of residence. The patterns of participation would be a common effect of sex and age, however, even if it was conditioned this would not change the adjustment set suggested for the DAG. A double-sided broken arrow between sex and race was included to represent the hypothesis of an interaction between race and sex in oral health.

The independence implications suggested by the DAG were assessed using statistical tests according to the nature of the dependent variable (e.g., ordinal logistic regression for ordinal outcome; linear regression for continuous variable). The level of significance to reject the testable implications was adjusted according to the *Holm-Bonferroni* method. The resulting DAG (Fig 1) had seven testable implications of independence, for which 19 tests were performed, considering that some variables were polytomous (adjusted significance level: 0.0026).

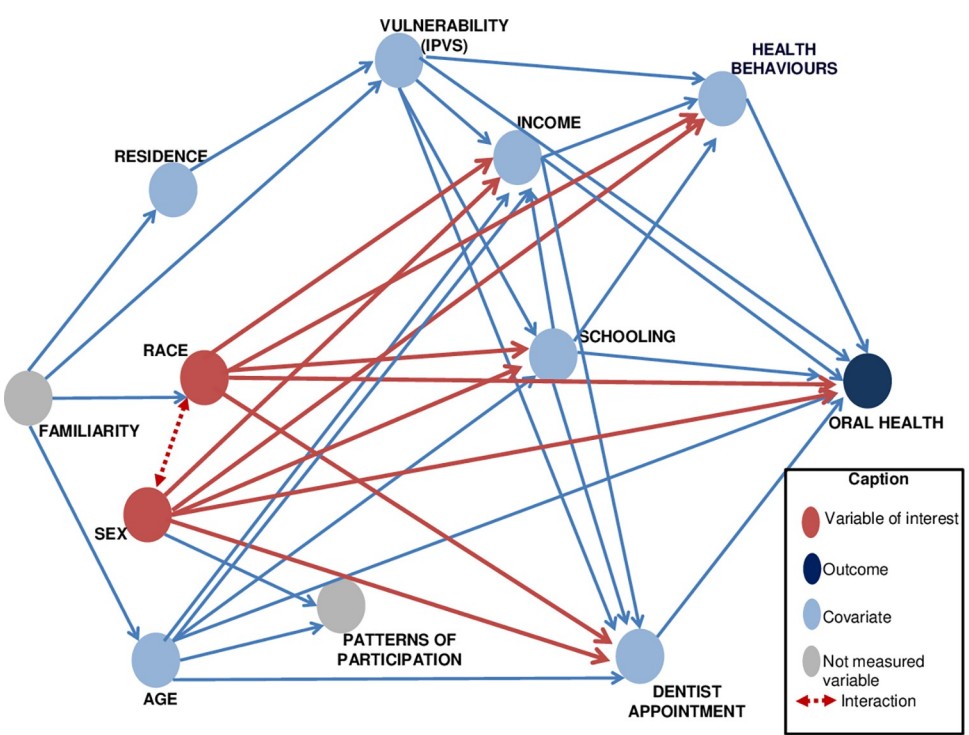

**Fig 1. Directed acyclic graphs, with the variables and implications tested, for tooth loss according to race/skin color and sex.**

However, none of the implications were rejected (p>0.10 for all tests), which was interpreted as an indicator of consistency between the DAG and the data [39, 45].

The minimal sufficient adjustment set suggested by the DAG included IPVS and Age. Consequently, adjustment for the unmeasured variables was not necessary. For the variables used, the most functional way to represent their association with the outcome was evaluated. Age was categorized by decile and IPVS remained with the categories of origin. Both variables were consistent and progressively associated with tooth loss (Fig 2), and therefore, included in the model as continuous.

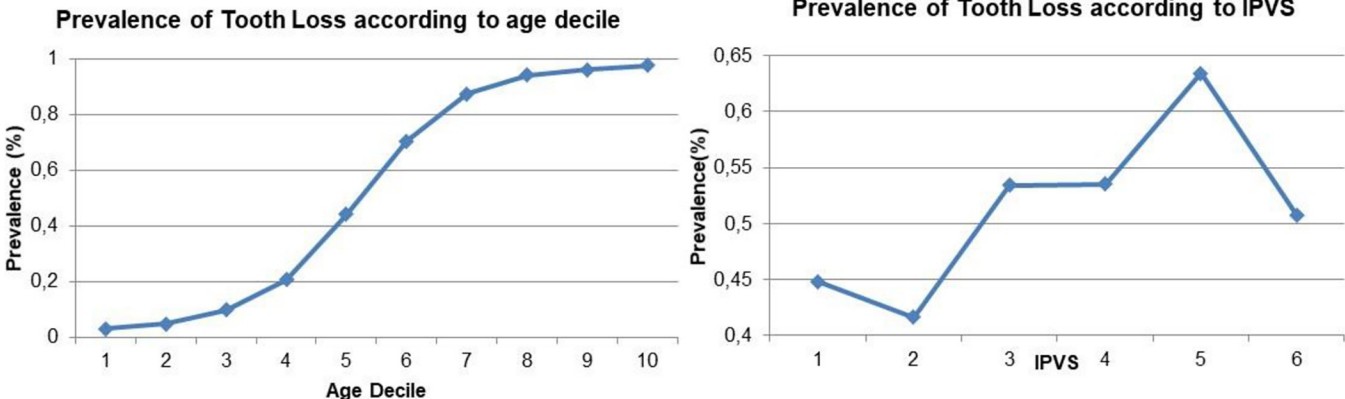

**Fig 2. Prevalence of tooth loss by age decile and by IPVS category.**

The analyses were conducted using the svy (survey) option of STATA 14.0, which considers the study design weights with complex sampling. In univariable analysis, associations between sociodemographic factors and race were tested using Pearson's chi-square test, according to sex. The adjusted Prevalence Ratio (PR) and their 95% intervals (95%CI) of tooth loss associations with the variables sex, race, and their interaction were estimated by Poisson regression (robust). Since this is a cross-sectional study, interaction measures were analogous to the Ratio of Relative Risks (RR) and Relative Excess Risk due to Interaction (RERI), which in our study we named Relative Prevalence Ratio (RPR) and Relative Excess Prevalence due to Interaction (REPI) on the multiplicative and additive scale, respectively. These estimates were adjusted by IPVS and Age. The interaction was evaluated on the additive and multiplicative scales, estimating, respectively, the relative excess prevalence due to interaction (REPI), estimated in a similar way to the relative excess risk due to interaction (RERI) calculated in cohort studies, and Relative Prevalence Ratio (RPR, analogous to the ratio of Relative Risks) and their 95%CI [40]. Considering that the event studied occurred after the exposures of interest (race and sex), the prevalence ratio was considered a good approximation of the RR. The results of the interaction analyses were presented according to the recommendations of Knol and VanderWeele [39]. As complementary measures to assess the interaction on the additive scale, we calculated the S-index and the attributable proportion (AP). However, because the denominator of the S-index was negative, of those both, we only reported the AP [48], which corresponds to the ratio between the REPI and the prevalence ratio of the category considered as doubly exposed (black women) [48].

To illustrate the interpretation of the resulting model, the predicted prevalence for each of the four categories defined by race and sex were calculated, taking as a reference the white man and a central tendency value of the adjustment variables.

The distribution of the tooth loss exhibited a significant number of "zeros" (see S1 Fig). Therefore, we decide to focus the primary analysis on the dichotomous outcome. However, to rule out that this decision would bias the conclusions due to loss of information, we also analyzed the outcome as the ordinal variable of tooth loss as aforementioned, to assess the consistency of the trends observed. Consequently, adjusted Odds Ratio were also estimated using multiple ordinal logistic regression models. In this case, four categories of sex and race combinations were created, and the model was also adjusted by age and IPVS. The project was approved by the Research Ethics Committee of the University of Campinas (Opinion no. 3744551/2019 of 04/12/2019; CAAE no. 24860219.4.0000.5404).

## Results

After excluding individuals who declared themselves to be indigenous, yellow individuals and another race, the data of 2962 people aged 10 years or more were analyzed. 558 (18.9%) black and 1060 (35.8%) white women took part in the research. Among men, 504 (17.0%) were black and 840 (28.4%), white. There was a trend in the black population for the younger age groups, while for whites this trend occurred for the older age groups. The percentage of women who receive an income greater than 3 MW was more than three times higher for white women than for black women. For schooling, the percentage who studied 9 years or more was 30% higher for white women. Black men also have lower income compared to white ones. The percentage of men living in a place of very low vulnerability was 10 times higher among white men and 3 times higher among black women, when comparing data between categories. (S1 Table).

It was observed that 52% had lost at least one or more teeth, in the total population studied. In the crude analysis, tooth loss of all teeth was more frequent in women, particularly in white

**Table 1. Adjusted Prevalence Ratios (PRs) for tooth loss comparing sex and race groups (ISACamp 2014/15).**

| | Tooth Loss | | | | PRs (95% CI) for race within strata of sex |
|---|---|---|---|---|---|
| | White Individuals | | Black Individuals | | |
| | Cases/ non-cases (Prevalence; IC95%) | PR (95%CI) | Cases/ non-cases (Prevalence; IC95%) | PR (95%CI) | |
| **Men** | 439/400 | **1** | 206/297 | 1.04 (0.88–1.23) | 1.04 (0.88–1.23) |
| | (52.3%; 48.9%-55.8%) | | (41%; 36.6%-45.4%) | p = 0.61 | p = 0.61 |
| **Women** | 597/462 | 0.94 (0.84–1.06) | 297/261 | **1.19 (1.05–1.34)** | **1.26 (1.13–1.42)** |
| | (56.4%; 53.3%-59.4%) | p = (0.32) | (53.2%; 49%-57.5%) | **p = 0.006** | **p < 0.001** |
| **PRs (95% CI) for sex within strata of race** | | 0.94 (0.84–1.06) | | **1.14 (1.02–1.27)** | |
| | | p = 0.32 | | **p = 0.021** | |

REPI (95%CI) = 0.20 (0.04–0.37) p = 0.02.

Relative Prevalence Ratio (RPR) (95%CI) = 1.21 (1.01–1.44); p = 0.04.

PRs are adjusted by IPVS and age, and weighted according to complex sample design.

women, and less frequent in black men (S1 Fig). After adjustments, the prevalence of tooth loss was 19% higher in black women compared to white men (PR: 1.19; 95%CI: 1.05–1.34). Among the female population, this event was 26% higher in black women compared to white women (PR: 1.26; 95%CI: 1.13–1.42). In the corresponding race category, black women had a 14% higher prevalence of tooth loss (PR: 1.14; 95%CI: 1.02–1.27) compared to black men (Table 1).

Thus, race and sex presented an interaction both on the additive and on the multiplicative scale (Table 1). In black women, the prevalence of tooth loss was 20% higher than expected (95%CI: 0.04–0.37; p = 0.02) on the additive scale; and 21% higher than expected (RPR = 1.21; 95%CI: 1.01–1.44; p = 0.04) on the multiplicative scale. According to the AP, 17.17% (95%CI: 2.5% - 31.86%, p = 0.02) of the prevalence in black women is due to the interaction. Fig 3 shows the estimated values for each of the categories defined by race and sex, taking as mean parameters of IPVS equal to 2 and age of 37 years. With these parameters, the tooth loss calculated according to the adjusted PR for black women was 45.1% (95%CI: 40.2%– 50.0%), which was higher than the values expected on the additive, 37.4% (95%CI: 29.8%– 44.7%) and multiplicative scales, 37.3% (95%CI: 29.7%– 45.0%).

Using ordinal logistic regression comparing tooth loss across the races and sex groups, adjusted for age and IPVS, it was observed that black women have twice as much odds of passing to a category of higher tooth loss (OR = 2.10: 95%CI = 1.52–2.91: p<0.001), compared to white men (Table 2). In this model, no other category was significantly different from the group of white men.

## Discussion

We have shown an interaction between sex and race on the risk for tooth loss on both the additive and the multiplicative scales. This result pointed to a higher prevalence of tooth loss among black women. The analysis of the interaction between race and sex in tooth loss was investigated for the first time on a based-population approach. In this study, information on toothlessness were collected by self-reports; however, self-reported tooth counting has proven to be a validated and reliable indicator of tooth loss [49, 50].

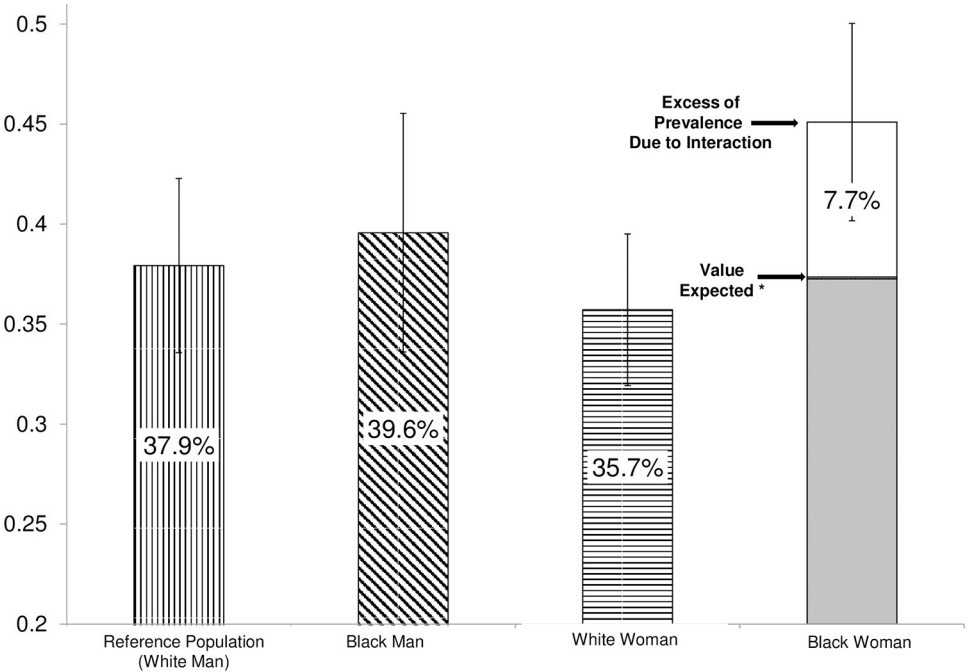

**Fig 3. Percentage of tooth loss estimated according to the prevalence ratio versus expected on the multiplicative and additive scales, according to the categories of race and sex adjusted by IPVS, ISACamp 2014/15.**

Some studies have found greater tooth loss in women [51–53]. According to a national epidemiological survey [54], between 15 and 19 years old, tooth loss is 20% in females and 13% in males. In adulthood, between 35 and 44 years old, estimates point to 21% of tooth loss in women and 18% in men. In the elderly, the percentage of edentulism rises to 56% in women, against 49% in men. Other authors have also found that the oral health condition in elderly women is worse compared to that of men [31].

Although there is a consensus on the disadvantage of women in oral health, information indicates that the female population seeks dental care more than men [11, 55], as it also happens for other types of health care [11]. This fact may be due to the greater self-perception of health that women have compared to men, in addition to the trend towards greater health care [4, 55]. However, there is evidence that the increased demand for dental treatment does not necessarily reflect better oral health conditions in women [52].

The findings of this study corroborate the results of a study carried out in Brazil, which evaluated the relationship between black individuals and tooth loss [56]. Although the mentioned study refers to the elderly population, the authors found that race is a limiting factor in

**Table 2. Odds ratio of tooth loss for the oral health outcome between sexes and race in the population with the IPVS and age conditions, ISACamp 2014/15.**

| Characteristic related to Oral Health (Tooth Loss) | OR[a,b] (95%CI) | P-value |
|---|---|---|
| **Black Men** | 1.15(0.78–1.70) | 0.49 |
| **White Women** | 0.92(0.69–1.22) | 0.56 |
| **Black Women** | **2.10(1.52–2.91)** | <0.001 |

[a] White Men was the reference category.

[b] Ordinal Logistic Regression adjusted by IPVS and Age.

access to oral health services, with black people being twice as likely to having never gone to the dentist compared to white people. In another Brazilian study, Souza et al. [57] found that, even after adjustments for income and living conditions, self-assessment of oral health, of the care received in dental consultations as bad, and consultation with the aim of extraction are more prevalent oral health issues in black people compared to white. The authors discuss issues of racial segregation beyond socioeconomic status.

Some studies [13, 58] lead to the discussion that being black individuals is a factor that interferes in the decision-making process to recommend tooth extraction. In addition, blacks suffered more posterior teeth extraction when compared to other ethnic groups [24]. It is possible that the dentist's decision has a pro-white bias to recommend a more complex treatment, more conservative of the tooth, while for black patients in the same condition, extraction is recommended [13, 59]. These tooth extraction practices may also be due to the lower socioeconomic level among the black population, which makes treatments to preserve teeth less accessible. Another explanation for radical decisions would be the issues of prejudice and discrimination, as has been discussed in other studies [13, 57, 58].

A possible explanation for the interaction between race and sex in tooth loss, with a disadvantage for black women, would be the health service conditions with a greater tendency to tooth extraction in black people [13, 58]. If the search and frequency of oral health services are higher among the female population [55], the burden of mutilating practices comes on the black woman.

Although progress has been made to reduce socioeconomic inequalities in the provision of public health services, it is worth noting that in oral health these disparities still tend to persist [23]. Currently, Brazil has some affirmative action policies, such as the quota system, in universities and in public tendering, to repair injustices against black individuals, indigenous, and the poor people. In addition, the National Policy for the Comprehensive Health of the Black Population was created to fight institutional racism and ethnic-racial prejudice in health institutions, instructing social movements, professionals, and health managers to provide better quality health care for the black population [60]. However, what can be verified is that these isolated policies are not able to avoid culturally rooted racism, especially when it comes to oral health services, which are typically for the elite.

Before these results, it is worth noting the need for public policies with affirmative actions and educational measures that can benefit black women to be maintained and rethought, to ensure universal and equal access, as advocated by the Single Health System (In Brazil, called SUS). In addition, this study also draws attention to the need for further studies on racial and sex disparities in oral health and dental issues occupying more space in government policies, to minimize the losses caused by difficulties and inequities.

The strengths of the study come from the novelty in evaluating the interaction between race and sex in tooth loss. Moreover it is a population-based study obtaining a probabilistic sample designed to represent a municipality with around one million inhabitants, with a questionnaire that allows the measurement of relevant variables for the control by potential confounding phenomena. In addition, a careful process of variable selections was made to control confounding. The DAG represented the structures of causal phenomena, which helped to choose the set of minimum variables for valid adjustment. Thus, it avoided unnecessary adjustments overfitting and collision biases that would affect the validity of the estimates. Thus, the association measures found, despite the design limitation, can be considered robust to describe the relationship between sociodemographic factors and to estimate the interaction between race and sex on the outcome of tooth loss.

The limitation of the study are that tooth loss was coded according to the question of the survey, and it was not possible to use the international classification of functional dentition.

However, regarding the categories analyzed, as described in S1 Fig, the outcome had an important proportion of people without tooth loss suggesting that a valid and informative approach is to focus on differentiating loss and no loss. Moreover, we also considered the outcome as an ordinal variable resulting in consistent associations. It is also worth considering the survival bias, since we are dealing with tooth loss; adult and elderly individuals are more exposed to this condition. Therefore, this bias could affect the similarity between the prevalence ratio and the relative risk. However, the prevalence measure obtained portrays the current disease burden and may indicate inequities in health between the groups analyzed.

## Conclusion

This study found a significant interaction between race and sex in tooth loss, with a disadvantage for black women. The female population who that self declared black was twice odds of having a worse dental condition than white men. The results could indicate a trend in oral health care based on mutilating decision-making in black individuals, affecting, especially, black women. Our findings suggest attention to the distribution pattern of tooth loss by race and sex and more studies can confirm the consistency between the interactions that indicate greater vulnerability in black women. Thus, after the evidence found in this work, it is recommended reinforce the surveillance to recognize and eliminate inequities in oral health care.

## Supporting information

**S1 Table. Sociodemographic characteristics according to sex and race, in the population aged 10 or over, ISACamp 2014–2015.**
(DOCX)

**S1 Fig. Percentage of frequency of tooth loss, according to race, by sex, ISACamp 2014/15.**
(DOCX)

## Author Contributions

**Conceptualization:** Lívia Helena Terra e Souza, Fredi Alexander Diaz-Quijano, Marilisa Berti de Azevedo Barros, Margareth Guimarães Lima.

**Data curation:** Lívia Helena Terra e Souza, Fredi Alexander Diaz-Quijano, Marilisa Berti de Azevedo Barros, Margareth Guimarães Lima.

**Formal analysis:** Lívia Helena Terra e Souza, Fredi Alexander Diaz-Quijano, Marilisa Berti de Azevedo Barros, Margareth Guimarães Lima.

**Funding acquisition:** Lívia Helena Terra e Souza, Fredi Alexander Diaz-Quijano, Marilisa Berti de Azevedo Barros, Margareth Guimarães Lima.

**Investigation:** Lívia Helena Terra e Souza, Fredi Alexander Diaz-Quijano, Marilisa Berti de Azevedo Barros, Margareth Guimarães Lima.

**Methodology:** Lívia Helena Terra e Souza, Fredi Alexander Diaz-Quijano, Marilisa Berti de Azevedo Barros, Margareth Guimarães Lima.

**Project administration:** Lívia Helena Terra e Souza, Fredi Alexander Diaz-Quijano, Marilisa Berti de Azevedo Barros, Margareth Guimarães Lima.

**Resources:** Lívia Helena Terra e Souza, Fredi Alexander Diaz-Quijano, Marilisa Berti de Azevedo Barros, Margareth Guimarães Lima.

**Software:** Lívia Helena Terra e Souza, Fredi Alexander Diaz-Quijano, Marilisa Berti de Azevedo Barros, Margareth Guimarães Lima.

**Supervision:** Lívia Helena Terra e Souza, Fredi Alexander Diaz-Quijano, Marilisa Berti de Azevedo Barros, Margareth Guimarães Lima.

**Validation:** Lívia Helena Terra e Souza, Fredi Alexander Diaz-Quijano, Marilisa Berti de Azevedo Barros, Margareth Guimarães Lima.

**Visualization:** Lívia Helena Terra e Souza, Fredi Alexander Diaz-Quijano, Marilisa Berti de Azevedo Barros, Margareth Guimarães Lima.

**Writing – original draft:** Lívia Helena Terra e Souza, Fredi Alexander Diaz-Quijano, Marilisa Berti de Azevedo Barros, Margareth Guimarães Lima.

**Writing – review & editing:** Lívia Helena Terra e Souza, Fredi Alexander Diaz-Quijano, Marilisa Berti de Azevedo Barros, Margareth Guimarães Lima.

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
