## [Decision Letter · Decision Letter 0]

13 Jan 2022

PONE-D-21-32428Race (black-white) and sex inequalities in tooth loss: a population-based studyPLOS ONE

Dear Dr. TERRA e SOUZA,

Thank you for submitting your manuscript to PLOS ONE. After careful consideration, we feel that it has merit but does not fully meet PLOS ONE’s publication criteria as it currently stands. Therefore, we invite you to submit a revised version of the manuscript that addresses the points raised during the review process.

ACADEMIC EDITOR:

Dear authors,

Three reviewers and I have reviewed the manuscript. Indeed, the manuscript explores a very interesting and relevant aspect in Dentistry and health care. Although the topic was relevant, some shortcomings and flaws were identified regarding appropriateness in exploring racial issues and analyses (outcomes choice and definition, and other statistical details). More detailed comments are addressed in reviewers reports and should be considered when revising the submission. They may be beneficial in clarifying some points and justifying others and maybe fundamental to making the manuscript suitable for publication in PlosOne.

We look forward to receiving your revised manuscript.

Kind regards,

Mariana Minatel Braga

Academic Editor

PLOS ONE

Journal Requirements:

Reviewers' comments:

Reviewer's Responses to Questions

**Comments to the Author**

1. Is the manuscript technically sound, and do the data support the conclusions?

Reviewer #1: No

Reviewer #2: Yes

Reviewer #3: Yes

2. Has the statistical analysis been performed appropriately and rigorously? 

Reviewer #1: Yes

Reviewer #2: Yes

Reviewer #3: Yes

3. Have the authors made all data underlying the findings in their manuscript fully available?

Reviewer #1: No

Reviewer #2: Yes

Reviewer #3: Yes

4. Is the manuscript presented in an intelligible fashion and written in standard English?

Reviewer #1: No

Reviewer #2: Yes

Reviewer #3: Yes

5. Review Comments to the Author

Reviewer #1: Revision PONE-D-21-32428

Race (black-white) and sex inequalities in tooth loss: a population-based study

1. The authors do not explain the concept of intersectionality in the introduction and do not justify the use of race in biomedical publications. Authors are based on too old references on the topic.

2. The RERI analysis is sound. However, without a proper justification for the outcome ( lost of 1 tooth or no – What is the usefulness? ) and a miscellaneous in ordinal category ( No loss, loss of 1 tooth, loss of more than one tooth, or loss of all teeth) in complicated to readers understand.

3. The DAD used by the authors is difficult to understand. Authors do not put "Place of residence" or "vulnerability index" linked to race. This is a mistake due to the structural factors related to ethnic inequalities. Recent evidence explaining racial inequities put it on evidence (neighbourhood conditions) in oral health (https://pubmed.ncbi.nlm.nih.gov/29634429/). Moreover, the authors did not analyze health behaviours, important mediators explaining tooth loss.

4. The outcome of tooth loss is very strange for me. For population comparisons, the authors do not use international classifications like functional dentition (more than 20 permanent teeth) and severe tooth loss ( less than ten remaining teeth).

5. I do not understand why authors opted to maintain ten years (children with mixed dentition) or more in the sample. Joining children, adolescents, adults and older adults in the same analysis. Difficult to interpret

6. Descriptive analysis need the n and CI 95% ( supplemental file 1)

7. How was tooth loss collected in the sample? By the number of teeth? Self-report one or more than one? Need more explanation…

8. Need more information on how authors collected data. How many examiners? Calibration? Provide detailed information.

9. Did the author proceed to sensitivity analysis? "It was verified that the categories of black and brown presented a similar association for tooth loss, in each sex" …and age between browns and blacks? Were they similar or different from Whites?

10. Why did the RERI analysis not report the attributable proportion [AP] and synergy index [S]?

11. Tables need to show the n of each category and prevalence and 95% CI, as recommended by Knol & VanderWeele (2012). This will add additional transparency for readers

12. Analyzing absolute prevalence in tooth loss, the percentages (Table 1 ) for white men is 52% ( authors did not put 95% CI), for white women 59% ( why prevalence ratio (PR)under 1?), black man 41% ( lower than white man? ) and black women 53% ( lower than white woman?). ….with slight absolute differences. Problems with the outcome selection as commented in comments #2 and #5 joining children with older adults…

Reviewer #2: The topic studied is relevant and well-justified in the Introduction section.

Methods are well-reported and properly designed to answer the research question. A few typos should be revised (e.g., Ln. 92 – the classification “very low” is presented twice).

Data were analyzed adequately, and the results are clearly presented and discussed.

Ln. 178 – Supplementary Figure 1 shows the opposite behavior.

Reviewer #3: I would like to congratulate the authors for their work and willingness to investigate differences between racial minorities and their privileged peers in an outcome that is of great importance in the field of public health. I'm sure the authors will appreciate my comments aimed at providing an outside view on how this manuscript can improve.

Introduction

- I would like to ask authors not to use "Blacks". Please use these words only as adjectives, using them as a noun is dehumanizing. Please write "Black individuals", for example.

- In the introduction, the authors present that the main health inequities related to race are: a) due to lesser access to health services; and b) due to socioeconomic issues. However, these are just two of the many factors where race is a cause of health inequities. It is not correct to reduce or omit other factors such as the slavery past of Brazilian society and Latin America. Thus, considering that the authors aim to address the influence of race on tooth loss, it is necessary that the authors deepen the discussion on this topic in the introduction.

- Please provide a very specific reason for what race means and why it is important to study it in the breakdown of oral health outcomes.

- I suggest that the authors review the objectives of the study, as in the format it is written, the authors state that there is an association between race and tooth loss. I suggest splitting it into two objectives: a) to investigate a possible association between race and tooth loss and b) assess possible interactions between race and sex in association with tooth loss.

There is also a need for greater attention to standardize the words used throughout the text, such as “race”, “race/skin color”, and “self-reported skin color” to refer to different racial groups in Brazil. In addition to needing to clearly answer, right in the introduction, why to study racial and gender inequities in face of tooth loss, the authors need to review the use of “skin color” throughout the text. If the authors are referring to the Brazilian classification system, it is important to make this clear, and to recognize first that the official wording has been "color/race" since 1991, and probably the same one used in the study. It should be clear that the complexity of the classification is beyond skin color and much more related to how society treats racial minorities, given racism and racial discrimination. I suggest that authors choose only "race" to facilitate understanding by international readers, making this reduction clear from the official classification. Please provide a very specific reason for what race means and why it is important to study it in the breakdown of oral health outcomes.

- The authors were not able to clearly show in the introduction how the current manuscript has originality compared to other articles already published.

- In addition, after reading the introduction, I would like to suggest some references for the authors to read before proceeding with the reformulation of the manuscript:

- Phelan JC, Link BG. Is Racism a Fundamental Cause of Inequalities in Health? Annu Rev Sociol. 2015;41(1):311-330.

- Bastos JL, Celeste RK, Paradies YC. Racial Inequalities in Oral Health. J Dent Res. 2018:22034518768536.

- Costa F,et al., Racial and regional inequalities of dental pain in adolescents: Brazilian National Survey of School Health (PeNSE), 2009 to 2015. 2021 Cad Saude Publica. 2021 Jun 25;37(6):e00108620.

- Williams DR, Priest N, Anderson NB. Understanding associations among race, socioeconomic status, and health: Patterns and prospects. Health Psychol. 2016;35(4):407-411.

Methods:

- In the section of study variables, please, provide all cathegorization of all variables. For example, In which category were yellow and indigenous individuals included?

- I recommend changing the label from the "black" to "black/brown individuals" since brown individuals were also included in this category.

- Please also present a paragraph supporting the assumptions presented in the DAG.

- how was tooth loss originally collected by DMF-T?

Discussion:

- Line 265, page 10. “SUS” needs to be written out in full.

6. PLOS authors have the option to publish the peer review history of their article (what does this mean?). If published, this will include your full peer review and any attached files.

Reviewer #1: No

Reviewer #2: No

Reviewer #3: **Yes: **Luiz Alexandre Chisini

---

## [Author Response · Author response to Decision Letter 0]

12 Mar 2022

PONE-D-21-32428

Race (black-white) and sex inequalities in tooth loss: a population-based study

PLOS ONE

Dear editors

We are submitting the new version of our manuscript. We would like to thank the reviewers for the suggestions, which have certainly enriched our paper. All comments and suggestions were fully taken into account in the preparation of this new version. Our point-by-point responses to the issues raise are listed below:

Review Comments to the Author

Reviewer #1: 

1. The authors do not explain the concept of intersectionality in the introduction and do not justify the use of race in biomedical publications. Authors are based on too old references on the topic.

Response: We modified the text and added the concept of intersectionality in the introduction: “In this way, it is worth considering the analytical strategy of intersectionality, developed by Crenshaw in 1989 to study the black feminism. Intersectionality, drawing on a set of social issues and perspectives, explain an experience of multiple subordination that cannot be reduced to the simple sum of underprivileged groups [34]”. (Lines 59-63).

We also justify the use of race in biomedical publications: “Racial minorities, in this case black people, may biologically embody the effects of racism [7,12], with everyday (or even less common) discriminatory exposures [12]. Adversities throughout life, such as poverty, psychosocial stresses, stereotypes, and the context of housing, can affect the physical and mental health, altering cardiocirculatory, metabolic and immunological functions [8]. In oral health is possible that inequities are due to poverty [10], levels of education [4] or discrimination in health care [13]. In Brazil, skin color has been studied as proxy of race and this indicator is commonly referred as color/race [14,15,16]. In health, in contrast only to the biologic approach, race may be considered as a concept socially constructed by historic dynamics and power relations [8,12].” (Lines 27-34)

2. The RERI analysis is sound. However, without a proper justification for the outcome (lost of 1 tooth or no – What is the usefulness? ) and a miscellaneous in ordinal category (No loss, loss of 1 tooth, loss of more than one tooth, or loss of all teeth) in complicated to readers understand.

Response: Thank you for the comment and I take the opportunity to clarify some points. We added this justification on the text: “Tooth loss is an important marker of oral health, since each tooth is a dental organ and its subtraction has consequences on occlusal adjustment, temporomandibular joint disorder, reduction of the masticatory board and consequently overload for the digestive system [37]. Besides, the loss of at least one tooth since it represents neglect in the dental field, resulting in an increase in the level of severity of oral diseases, in addition to reflecting the model of oral health care adopted and the way individuals understand the condition [38].” (Line 97-103)

Moreover, in the last paragraph of methodology, we complemented: “… Therefore, we decided to focus the primary analysis on the dichotomous outcome, which also represents the most practical way to assess and present interactions [39, 40]. However, on the other hand, the analysis of the same variable as ordinal can show us a gradient of tooth loss, varying its degrees of loss [41]. This analysis was used only to confirm trends in tooth loss in the black population. (Lines 103-106)

3. The DAD used by the authors is difficult to understand. Authors do not put "Place of residence" or "vulnerability index" linked to race. This is a mistake due to the structural factors related to ethnic inequalities. Recent evidence explaining racial inequities put it on evidence (neighbourhood conditions) in oral health (https://pubmed.ncbi.nlm.nih.gov/29634429/). Moreover, the authors did not analyze health behaviours, important mediators explaining tooth loss.

Response: We appreciate the comments and the reference indicated and we reviewed the DAG. We understand that neighborhood conditions are important aspects in the health and specifically in oral health. But our thinking about the DAG is the issues of determination. We understand that race is closely linked to places of residence, but these variables are not directly related in a causal diagram. Explicitly, a race does not cause a place of residence, nor a place of residence cause a race. These variables can be associated, as the reviewer correctly indicated, but the only explanation it would be sharing a common cause. 

Because can be a long list of potential unmeasured variables, fitting this role of being a common cause of place and race, we follow a common convention that visually simplifies the DAG by representing all unmeasured variables with the same causal structure (i.e., the same arrows in and out) as a single node (Digitale JC, Martin JN, Glymour MM. Tutorial on directed acyclic graphs. J Clin Epidemiol. 2021 Aug 8:S0895-4356(21)00240-7. doi: 10.1016/j.jclinepi.2021.08.001). In this case, we included the concept of “familiarity” as that node representing all common determinants of both, place of residence and race. 

In the same sense, vulnerability (as a context variable measured with the IPVS) in which people are born, would neither determine nor be determined by race. But it would be determined by factors such as familiarity and place of residence. 

On the other hand, we agree with reviewer regarding the health behaviours as important mediators of demographic determinants. Therefore, we included a variable on the DAG (health behavior) determined by sex, race and age, which would be causally connected with the outcome (Please see new DAG with this inclusion). However, such our objective was to estimate the total effects (not to analyse mediation) of race and sex, specifying their interaction, no additional adjustment was necessary. 

During revision of DAG, we perceived that pattern of participation would be better described as a common effect of sex and age than a common cause. So, as it can be conditionate by decision of participate of population, it can imply a collider bias, opening a path. However, this bias it would be also corrected by adjusting by age. Therefore, again, the set of variables indicated to adjusted remain the same. Inserted in the text lines 149-161.

4. The outcome of tooth loss is very strange for me. For population comparisons, the authors do not use international classifications like functional dentition (more than 20 permanent teeth) and severe tooth loss (less than ten remaining teeth).

Response: The question used to collect the tooth loss data used as answered in question #7 and added to the text used the categories (1) No, (2) yes, only one tooth, (3) yes, more than one tooth, (4) yes, all teeth, being used in this way in other studies by Gomes Filho and Verçosa. We insert an information on the studies limitations: “Also, tooth loss was coded according to question of the survey, and it was not possible to use the international classification of functional dentition.” (Lines 323-324).

5. I do not understand why authors opted to maintain ten years (children with mixed dentition) or more in the sample. Joining children, adolescents, adults and older adults in the same analysis. Difficult to interpret

Response: We considered important to analyze a wide age range, since we believe that inequality problems affect the population from early stages of life. In this sense, we observed that the associations were consistent and independent of age in the multiple model. We understand the reviewer's concern, however, as we described in the methods, our interviewers were trained and instructed to ask the question about tooth loss excluding teeth extracted for orthodontic reasons, disregarding teeth extracted for orthodontic, wisdom and baby tooth reasons (this information has been added to the text). This can minimize the risk of a classification bias that the age group would have in relation to studies of tooth loss and social conditions. (Lines 96-97).

6. Descriptive analysis need the n and CI 95% (supplemental file 1)

Response: Thanks for the suggestion, the data was entered into the table. (Page 17)

7. How was tooth loss collected in the sample? By the number of teeth? Self-report one or more than one? Need more explanation…

Response: We added a information in the text: “The outcome variable was self-referred tooth loss, extracted from the question: “Have you ever lost a tooth (upper or lower)? If so, did he lose one, more than one or all of his teeth? (disregard extracted teeth to place braces, wisdom and milk teeth)”, with the options to response: (1) No, (2) yes, only one tooth, (3) yes, more than one tooth, (4) yes, all teeth.” (Lines 90-93)

We also entered the information that the tooth loss was self-reported: “The analysis of the interaction between race and sex in tooth loss was investigated for the first time on a population basis. In this study, information on toothlessness were collected by self-reports…” (Lines 261-262)

8. Need more information on how authors collected data. How many examiners? Calibration? Provide detailed information.

Response: We added the following information: “Data were collected from 3021 people aged 10 or over [36], using a pre-coded questionnaire, applied through tablets, by interviewers trained." (Lines 85-87). “When asking about tooth loss, the interviewers informed the respondents to exclude teeth extracted for orthodontic reasons, disregarding teeth extracted for braces, wisdom and baby teeth.” (Lines 96-97)

9. Did the author proceed to sensitivity analysis? "It was verified that the categories of black and brown presented a similar association for tooth loss, in each sex" …and age between browns and blacks? Were they similar or different from Whites?

Response: We inform that it was found that the age categories for blacks and browns showed a similar association for tooth loss. This information was inserted in the text: “ It was verified that the categories of black and brown presented a similar association for tooth loss, in each sex and in each age, so these two categories were joined to compose the group of black individuals (Also used by IBGE-Brazilian Institute of Geography and Statistics) [42]. ” (Lines 110-112).

10. Why did the RERI analysis not report the attributable proportion [AP] and synergy index [S]?

Response: We appreciate the comment and added the suggested analyzes. In methodology, we included: “As complementary measures to assess the interaction on the additive scale, we calculated the S-index and the attributable proportion (AP). However, because the denominator of the S-index was negative, of those both, we only reported the AP [48], which corresponds to the ratio between the REPI and the prevalence ratio of the category considered as doubly exposed (black women) [48]”. (Lines 188-192). Moreover, in result section, we reported that “According to the AP, 17.17% (95%CI: 2.5% - 31.86%, p=0.02) of the prevalence in black women is due to the interaction”. (Lines 233-234).

11. Tables need to show the n of each category and prevalence and 95% CI, as recommended by Knol & VanderWeele (2012). This will add additional transparency for readers.

Response: We appreciate the recommendation. We reviewed the manuscript and verified that the table already followed exactly the orientations of the reference. That is, the number of cases and non-cases in each category, the association measure and its confidence interval. However, we agree that it would be good to report unadjusted prevalence. Therefore, we added the prevalence with its confidence interval in each category. (Page 9)

12. Analyzing absolute prevalence in tooth loss, the percentages (Table 1 ) for white men is 52% ( authors did not put 95% CI), for white women 59% ( why prevalence ratio (PR)under 1?), black man 41% ( lower than white man? ) and black women 53% ( lower than white woman?). ….with slight absolute differences. Problems with the outcome selection as commented in comments #2 and #5 joining children with older adults…

Response: Different than unadjusted measures, the estimates of prevalence ratios in this manuscript considered both the sample weights of the complex design of the study and the adjustment for age and IPVS, as informed in the text: “The analyses were conducted using the svy (survey) option of STATA 14.0, which considers the study design weights with complex sampling”. (Lines 177-178). We appreciate the comment, and we agree. So, we added the explanation on the footnote of the table: “PRs are adjusted by IPVS and age, and weighted according to complex sample design”

We reviewed the calculations and confirmed the values. (Page 9). 

Reviewer #2: 

The topic studied is relevant and well-justified in the Introduction section.

Methods are well-reported and properly designed to answer the research question. A few typos should be revised (e.g., Ln. 92 – the classification “very low” is presented twice).

Response: We appreciate the recognition and inform you that the suggestions have been modified in the text. (Line 129).

Data were analyzed adequately, and the results are clearly presented and discussed.

Ln. 178 – Supplementary Figure 1 shows the opposite behavior.

Response: In the crude analysis, black men have the lowest tooth loss and among women, black women have less tooth loss compared to white women. However, when we do the adjusted analysis, we find the opposite.

As explained in the text: “In the crude analysis, tooth loss of all teeth was more frequent in women, particularly in white women, and less frequent in black men (Supplementary Figure 1). After adjustments, the prevalence of tooth loss was 19% higher in black women compared to white men (PR: 1.19; 95%CI: 1.05-1.34).” (Lines 217-220)

Reviewer #3:

 I would like to congratulate the authors for their work and willingness to investigate differences between racial minorities and their privileged peers in an outcome that is of great importance in the field of public health. I'm sure the authors will appreciate my comments aimed at providing an outside view on how this manuscript can improve.

Response: We appreciate the recognition.

Introduction

1. I would like to ask authors not to use "Blacks". Please use these words only as adjectives, using them as a noun is dehumanizing. Please write "Black individuals", for example.

Response: We changed for “black individuals”.

2. In the introduction, the authors present that the main health inequities related to race are: a) due to lesser access to health services; and b) due to socioeconomic issues. However, these are just two of the many factors where race is a cause of health inequities. It is not correct to reduce or omit other factors such as the slavery past of Brazilian society and Latin America. Thus, considering that the authors aim to address the influence of race on tooth loss, it is necessary that the authors deepen the discussion on this topic in the introduction. 

Response: We appreciate the suggestion and inform you that we have modified the text as suggested. (Lines 25-34).

3. Please provide a very specific reason for what race means and why it is important to study it in the breakdown of oral health outcomes.

Response: Thank you. We have included this information in this paragraph of the text: “Racial minorities, in this case black people, may biologically embody the effects of racism [7,12], with everyday (or even less common) discriminatory exposures [12]. Adversities throughout life, such as poverty, psychosocial stresses, stereotypes, and the context of housing, can affect the physical and mental health, altering cardiocirculatory, metabolic and immunological functions [8]. In oral health is possible that inequities are due to poverty [10], levels of education [4] or discrimination in health care [13]. In Brazil, skin color has been studied as proxy of race and this indicator is commonly referred as color/race [14,15,16]. In health, in contrast only to the biologic approach, race may be considered as a concept socially constructed by historic dynamics and power relations [8,12].”

4. I suggest that the authors review the objectives of the study, as in the format it is written, the authors state that there is an association between race and tooth loss. I suggest splitting it into two objectives: a) to investigate a possible association between race and tooth loss and b) assess possible interactions between race and sex in association with tooth loss.

Response: We accept the suggestion, and the text was changed lines 64-65.

5.There is also a need for greater attention to standardize the words used throughout the text, such as “race”, “race/skin color”, and “self-reported skin color” to refer to different racial groups in Brazil. In addition to needing to clearly answer, right in the introduction, why to study racial and gender inequities in face of tooth loss, the authors need to review the use of “skin color” throughout the text. If the authors are referring to the Brazilian classification system, it is important to make this clear, and to recognize first that the official wording has been "color/race" since 1991, and probably the same one used in the study. It should be clear that the complexity of the classification is beyond skin color and much more related to how society treats racial minorities, given racism and racial discrimination. I suggest that authors choose only "race" to facilitate understanding by international readers, making this reduction clear from the official classification. Please provide a very specific reason for what race means and why it is important to study it in the breakdown of oral health outcomes.

Response: Thank you. We used “race” to define “color/race” and we clarify in the methods section (lines 107-110): “The race variable studied was self-reported. In Brazil this is the official method of racial classification since 1991, based on the individuals’ own perception of the color of their skin. In the present study, race/skin color variable was denominated “race” and categorized as black, brown, and white individuals.”. 

In the introduction, we also added an information of use of race as a social construct: “In health, in contrast only to the biologic approach, race may be considered as a concept socially constructed by historic dynamics and power relations [8,12].” (Lines 33-34)

After a paragraph explaining about gender inequalities in tooth loss (lines 59-63)., we include the need and possibility of studying race and gender in intersectional aspects: “The evidence of racial and gender inequalities in tooth loss could bring the subgroup of black women underprivileged in relation to this theme. In this way, it is worth considering the analytical strategy of intersectionality, developed by Crenshaw in 1989 to study the black feminism. Intersectionality, drawing on a set of social issues and perspectives, explain an experience of multiple subordination that cannot be reduced to the simple sum of underprivileged groups [34].”

6. The authors were not able to clearly show in the introduction how the current manuscript has originality compared to other articles already published.

Response: We thank you for the note made, to make the originality of the study clear, we added: “To our knowledge, there are no studies looking at the relationship between race and sex in relation to tooth loss and this study contributes to filling this gap.” (Lines 66-67).

7. In addition, after reading the introduction, I would like to suggest some references for the authors to read before proceeding with the reformulation of the manuscript:

-rdszz Phelan JC, Link BG. Is Racism a Fundamental Cause of Inequalities in Health? Annu Rev Sociol. 2015;41(1):311-330.

- Bastos JL, Celeste RK, Paradies YC. Racial Inequalities in Oral Health. J Dent Res. 2018:22034518768536.

- Costa F,et al., Racial and regional inequalities of dental pain in adolescents: Brazilian National Survey of School Health (PeNSE), 2009 to 2015. 2021 Cad Saude Publica. 2021 Jun 25;37(6):e00108620.

- Williams DR, Priest N, Anderson NB. Understanding associations among race, socioeconomic status, and health: Patterns and prospects. Health Psychol. 2016;35(4):407-411.

Response: We are very grateful for the suggestions and inform you that they were fundamental in making our introduction more grounded.

Methods:

8. In the section of study variables, please, provide all cathegorization of all variables. For example, In which category were yellow and indigenous individuals included?

Response: We added an information about yellow and indigenous individuals: “Individuals who declared themselves as yellow, indigenous, or other races were not studied, as they represented less than 2% of the population.” (Lines 113-114).

9. I recommend changing the label from the "black" to "black/brown individuals" since brown individuals were also included in this category.

Response: We made the change in the text. (Lines 110-111)

10. Please also present a paragraph supporting the assumptions presented in the DAG. 

Response: We made the change in the text line 146. “In the DAG, we represented the effects of race and sex on oral health, which can be mediated by income [3,4], education [4] and, more proximally to the outcome, by health behaviors and the dental care [6]. Other variables, such as age, familiarity, patterns of participation in the research, place of residence and IPVS [7], were also considered in the causal diagram (Figure 1). Because it can be a list of unmeasured common causes of both place and race, we follow a convention that visually simplifies the DAG by representing all unmeasured variables with the same causal structure (i.e., the same arrows in and out) as a single node (47). In this case, we included the concept of “familiarity” as that node representing all common determinants of both, place of residence and race. On the other hand, vulnerability (as a context variable measured with the IPVS) in which people are born, would neither determine or be determined by race. But it would be determined by factors such as familiarity and place of residence. The patterns of participation would be a common effect of sex and age, however, even if it was conditioned this would not chance the adjustment set suggested for the DAG. A double-sided broken arrow between sex and race was included to represent the hypothesis of an interaction between race and sex in oral health.” (Lines 149-161)

- how was tooth loss originally collected by DMF-T?

Response: We added an information about the original question about tooth loss: “The outcome variable was self-referred tooth loss, extracted from the question: “Have you ever lost a tooth (upper or lower)? If so, did he lose one, more than one or all of his teeth? (disregard extracted teeth to place braces, wisdom and milk teeth)”, with the options to response: (1) No, (2) yes, only one tooth, (3) yes, more than one tooth, (4) yes, all teeth.” (Lines 90-93)

We also entered the information that the tooth loss was self-reported: “The analysis of the interaction between race and sex in tooth loss was investigated for the first time on a population basis. In this study, information on toothlessness were collected by self-reports; ..” (Lines 261-262)

Discussion:

11. Line 265, page 10. “SUS” needs to be written out in full.

Response: We appreciate your suggestion and inform you that it has been added to the text line 306.

---

## [Decision Letter · Decision Letter 1]

8 Aug 2022

PONE-D-21-32428R1Race (black-white) and sex inequalities in tooth loss: a population-based studyPLOS ONE

Dear Dr. TERRA e SOUZA,

Thank you for submitting your manuscript to PLOS ONE. After careful consideration, we feel that it has merit but does not fully meet PLOS ONE’s publication criteria as it currently stands. Therefore, we invite you to submit a revised version of the manuscript that addresses the points raised during the review process.

ACADEMIC EDITOR: Although the authors have addressed most of the raised points in the present revised version, some aspects remained to be explored, as detailed by the reviewers. These aspects matter to important methodologic choices and findings interpretations and must be revised or discussed if the manuscript is published. Then, we recommend carefully revising them to gather a final acceptable version to be published in PlosOne.

We look forward to receiving your revised manuscript.

Kind regards,

Mariana Minatel Braga

Academic Editor

PLOS ONE

Reviewers' comments:

Reviewer's Responses to Questions

**Comments to the Author**

1. If the authors have adequately addressed your comments raised in a previous round of review and you feel that this manuscript is now acceptable for publication, you may indicate that here to bypass the “Comments to the Author” section, enter your conflict of interest statement in the “Confidential to Editor” section, and submit your "Accept" recommendation.

Reviewer #2: All comments have been addressed

Reviewer #3: All comments have been addressed

2. Is the manuscript technically sound, and do the data support the conclusions?

Reviewer #2: Yes

Reviewer #3: Yes

3. Has the statistical analysis been performed appropriately and rigorously? 

Reviewer #2: Yes

Reviewer #3: Yes

4. Have the authors made all data underlying the findings in their manuscript fully available?

Reviewer #2: No

Reviewer #3: Yes

5. Is the manuscript presented in an intelligible fashion and written in standard English?

Reviewer #2: Yes

Reviewer #3: Yes

6. Review Comments to the Author

Reviewer #2: All suggestions done by this reviewer in the first version were adequately addressed. Moreover, accepting the recommendations done by the other reviewer substantially improved the revised version of the manuscript.

Reviewer #3: The present study has improved considerably in this version. I still have a few points to point out.

Skin color self-report should not be a limitation of the study, as this is the best strategy to investigate race in Brazil, since race and skin color can be interpreted as synonyms in the Brazilian context.

In addition, rethinking from the statements of reviewer #1, I believe that the inclusion of individuals with mixed dentition is really difficult to justify and can imply large biases as well as the way of categorizing tooth loss. The authors mention that they used the question of the study by Gomes Filho and Verçosa, however, they collected the answers differently. This is a critical point of the study.

The authos relate “After adjustment for covariates suggested by a directed acyclic graph”; however, is not appropriate describe that DAG “suggest” adjustments. Report only that DAG was used to select covariates.

7. PLOS authors have the option to publish the peer review history of their article (what does this mean?). If published, this will include your full peer review and any attached files.

Reviewer #2: No

Reviewer #3: **Yes: **Luiz Alexandre Chisini

---

## [Author Response · Author response to Decision Letter 1]

16 Aug 2022

Dear editors

We are submitting the new version of our manuscript. We would like to thank the reviewers for the suggestions, which have certainly enriched our paper. All comments and suggestions were fully taken into account in the preparation of this new version. Our point-by-point responses to the issues raise are listed below:

Review Comments to the Author

1. If the authors have adequately addressed your comments raised in a previous round of review and you feel that this manuscript is now acceptable for publication, you may indicate that here to bypass the “Comments to the Author” section, enter your conflict of interest statement in the “Confidential to Editor” section, and submit your "Accept" recommendation.

Reviewer #2: All comments have been addressed

Reviewer #3: All comments have been addressed

Response: We appreciate the comments.

2. Is the manuscript technically sound, and do the data support the conclusions?

Reviewer #2: Yes

Reviewer #3: Yes

Response: We appreciate the comments.

3. Has the statistical analysis been performed appropriately and rigorously?

Reviewer #2: Yes

Reviewer #3: Yes

Response: We appreciate the comments.

4. Have the authors made all data underlying the findings in their manuscript fully available?

The PLOS Data policy requires authors to make all data underlying the findings described in their manuscript fully available without restriction, with rare exception (please refer to the Data Availability Statement in the manuscript PDF file). The data should be provided as part of the manuscript or its supporting information, or deposited to a public repository. For example, in addition to summary statistics, the data points behind means, medians and variance measures should be available. If there are restrictions on publicly sharing data e.g. participant privacy or use of data from a third party—those must be specified.

Reviewer #2: No

Reviewer #3: Yes

Response: The raw data and a codebook were deposited in the public repository as requested. Access address: https://doi.org/10.25824/redu/J9YOTW .

5. Is the manuscript presented in an intelligible fashion and written in standard English?

Reviewer #2: Yes

Reviewer #3: Yes

Response: We appreciate the comments.

6. Review Comments to the Author

Reviewer #2: All suggestions done by this reviewer in the first version were adequately addressed. Moreover, accepting the recommendations done by the other reviewer substantially improved the revised version of the manuscript.

Response: We appreciate the comments.

Reviewer #3: The present study has improved considerably in this version. I still have a few points to point out.

Skin color self-report should not be a limitation of the study, as this is the best strategy to investigate race in Brazil, since race and skin color can be interpreted as synonyms in the Brazilian context.

Response: We agree with the reviewer. We removed the self-declaration race/ skin color from the limitation.

In addition, rethinking from the statements of reviewer #1, I believe that the inclusion of individuals with mixed dentition is really difficult to justify and can imply large biases as well as the way of categorizing tooth loss. The authors mention that they used the question of the study by Gomes Filho and Verçosa, however, they collected the answers differently. This is a critical point of the study.

Response: The interviewers did not consider tooth loss those extracted for orthodontic reasons, disregarding teeth extracted for braces, wisdom, and milk (baby) teeth (lines 105-106). Further, regarding the categories analyzed, as described in figure S1, the outcome had an important proportion of people without tooth loss suggesting that a valid and informative approach is to focus on differentiating loss and no loss. Moreover, we also considered the outcome as an ordinal variable resulting in consistent associations (as commented in the discussion, lines 330-336). Therefore, no additional modifications were made regarding this point.

The authos relate “After adjustment for covariates suggested by a directed acyclic graph”; however, is not appropriate describe that DAG “suggest” adjustments. Report only that DAG was used to select covariates.

Response: We appreciate the suggestion has been modified as recommended. Line 18.

7. PLOS authors have the option to publish the peer review history of their article (what does this mean?). If published, this will include your full peer review and any attached files.

Do you want your identity to be public for this peer review? For information about this choice, including consent withdrawal, please see our Privacy Policy.

Reviewer #2: No

Reviewer #3: Yes: Luiz Alexandre Chisini

Response: We appreciate the comments.

---

## [Editor Report · Decision Letter 2]

29 Sep 2022

Race (black-white) and sex inequalities in tooth loss: a population-based study

PONE-D-21-32428R2

Dear Dr. TERRA e SOUZA,

We’re pleased to inform you that your manuscript has been judged scientifically suitable for publication and will be formally accepted for publication once it meets all outstanding technical requirements.

Kind regards,

Mariana Minatel Braga

Academic Editor

PLOS ONE

Additional Editor Comments (optional):

Considering the improvements in the revised version and/or adequate answers to reviewers' queries, this manuscript could be, at the present format, acceptable for publication. We appreciate authors' efforts in improving this final version of the manuscript.

---

## [Editor Report · Acceptance letter]

5 Oct 2022

PONE-D-21-32428R2 

Race (black-white) and sex inequalities in tooth loss: a population-based study 

Dear Dr. Terra e Souza:

I'm pleased to inform you that your manuscript has been deemed suitable for publication in PLOS ONE. Congratulations! Your manuscript is now with our production department. 

Kind regards, 

on behalf of

Dr. Mariana Minatel Braga 

Academic Editor

PLOS ONE